# The Positive Effects of Unneeded Consumption Behaviour on Consumers during the COVID-19 Pandemic

**DOI:** 10.3390/ijerph18126404

**Published:** 2021-06-13

**Authors:** Jianjia He, Shengmin Liu, Tingting Li, Thi Hoai Thuong Mai

**Affiliations:** 1Business School, University of Shanghai for Science and Technology, Shanghai 200093, China; liushengmin@usst.edu.cn (S.L.); leett1226@163.com (T.L.); alinathuong@163.com (T.H.T.M.); 2Super Network Research Center, University of Shanghai for Science and Technology, Shanghai 200093, China

**Keywords:** unneeded consumption behaviour, work engagement, recovery level, indulgence, perceived consumer effectiveness

## Abstract

The COVID-19 pandemic has become an important global contagion that requires workers to implement necessary behaviours to cope. Based on the conservation of resources theory, the present studies explore the effects of unneeded consumption behaviour on consumers’ recovery level and work engagement and the moderated mediating process of such relationships. Using a purchasing experiment, study 1 examined the positive effect of unneeded consumption behaviour on recovery among 100 MBA students. Using the experience sampling method, the data in study 2 were collected from 115 consumers (employees) using ten iterations of 2-day continual questionnaires (Sunday and the following Monday) during the COVID-19 pandemic. The results from multilevel structural equation modelling indicate that unneeded consumption behaviour positively impacts work engagement in a moderated mediating mode. Consumer indulgence positively moderates the mediating effect of recovery level on the relationship between indulgent consumption behaviour and work engagement, while perceived consumer effectiveness negatively moderates the mediating effect of recovery level. This paper also identifies the value of transformation from consumption to work during the COVID-19 pandemic.

## 1. Introduction

During the COVID-19 pandemic and community lockdown, most workers had to work from home, and their consumption choices were limited. At times, they would be permitted a brief amount of time to purchase necessary products and would often buy unnecessary items. Why have these phenomena spread in the nationwide lockdown during the COVID-19 pandemic?

Unneeded consumption is a kind of state when consumers buy more products than they require [1]. For example, one consumer needs forty pieces of bread to eat in 10 days; however, fifty pieces of bread is purchased even though the bread quality guarantee is ten days. Thus, ten pieces of bread are classed as unnecessary consumption. Unneeded consumption aims to satisfy psychological desires, where the main focus is on the consumption of material possessions to achieve the value of psychological well-being. Income, stress and consumption habit may influence the vary of unneeded consumption [2,3,4]. For example, consumers may neglect the quantities needed when they buy their favourite goods. A consumer wants to enjoy the purchasing process to relieve stress despite their budget and the necessary quantities required as high income and consumption inertia is in effect.

A similar topic that is a feature of crisis-related insecurity is panic buying and stockpiling behaviour, with such related behaviour being a widely reported response to COVID-19 intervention measures enacted by the government [5,6,7]. Understanding such purchasing and stockpiling behaviour is essential for the disaster management sector and retail organisations [8].

Unneeded consumption is different from panic buying. Panic buying is caused by the object of acquiring security and preparing for future needs [9]. Unneeded consumption is activated by the habitual trend or relieving stress after lockdown [3,4,5]. Panic buying is anchoring on the shelf of markets but unneeded consumption anchors on consumers’ basic needs. Panic buying shows consumers buy out of the shelf on the market but unneeded consumption only means that the quantities of goods they buy exceed consumers’ necessary quantities from basic needs. The direct factor of unneeded consumption may be pressed in lockdown for a long time and consumers need to relieve their pressed motive by engaging in unneeded consumption. Unneeded consumers do not consider whether or not the goods they buy is useful for the future, which is different from panic buying. Unneeded consumption happens with their habitual inertia or relieving their stress, not with planning intention to hoarding or stockpiling, which is the act of collecting and safeguarding a large number of possessions for future use [10].

Unneeded consumption of goods and services increases the use of natural resources that is a major cause of environmental problems, including global warming, polluted air and water and biodiversity reduction [11]. Unneeded consumption also results in producing more packaging material that must be disposed of. On the other hands, previous scholars have found many inhibitors to minimise redundant consumption [12]. Unnecessary consumption behaviours could be limited through the promotion of social responsibility or self-control trends [1]. Meanwhile, such behaviours positively influence psychological well-being or happiness [13,14]. More specifically, unneeded consumption in a home environment is seen as a tool to connect family members during the COVID-19 pandemic. According to Bahagia et al. [15], in this pandemic, most homemakers have to face many challenges such as supervising their children while doing house chores. Further, some may be working from home and considered an invisible pressure from the family aspect, therefore buying more than they need is a way to reduce stress during social distancing. However, why has such behaviour arisen during the COVID-19 pandemic? Does engaging in unneeded consumption behaviour have a positive effect on consumers (workers)?

Previous studies have focused on the detriments of unneeded consumption behaviour on the environment and demonstrated emotional regulation strategies used by consumers in surplus purchasing [14]. Specifically, surplus purchasing can be reappraised according to the emotions experienced in the purchasing process. For example, assuming that somebody likes bread and would like to buy one bread for their dinner. After the purchasing process, they can recognise that the quantity of bread they have purchased is more than their basic need. The reason is in the purchasing process: they will experience the positive emotion, as in the hedonic motivation of impulse buying [16], and reappraise the benefits of such behaviour, to experience positive emotions on the spot, which improves consumers’ recovery level [17,18]. If consumers experience high levels of recovery from unneeded consumption behaviour, what is the subsequent outcome?

During the COVID-19 pandemic, more consumers worked from home as both consumers and employees; they needed a way to recover from their psychological resources because of the burden of additional stressors such as individual isolation and fear of COVID-19, which made it difficult to engage in their work due to frequent interruptions in the form of community lockdowns and daily virus detection. If consumers experience positive emotions and recovery in unneeded consumption behaviour, would they direct these psychological benefits to their work roles and improve their engagement? If so, when would this shift occur?

To answer these questions, we built a moderated mediating model to explore the positive influence of unneeded consumption behaviour on work engagement via the shift of recovery level from life to the workplace. The COVID-19 pandemic has caused many psychological injuries for most people, and they need a way to replenish themselves. Individual coping and self-regulation are related to explaining individual responses or reactions to emerging stressors from the COVID-19 pandemic [19]. The conservation of resources theory (COR) can explain how these workers/consumers cope with these stressors (unneeded consumption behaviour-recover-engagement). Further, the individualism-collectivism cultural perspective was used to explore when unneeded consumption behaviour is positive.

Addressing the repairing function of indulgence [20] in the individualism value, we chose consumer indulgence as a positive mediator between unneeded consumption behaviour and work engagement. Based on the effects of responsibility reminders [1] in the collectivism value, perceived consumer effectiveness was considered as the negative mediator in the hypothetical model.

## 2. Theoretical Framework

The COVID-19 pandemic emerged as a traumatic event that required people to make sense of the situation and choose appropriate reactions. Culture plays an important role in shaping the way individuals assess or cope with stressors related to the COVID-19 pandemic. Further, there are huge cross-cultural differences in individuals’ appraisals of stressors, choices of coping strategies and indicators of adaptive outcomes [21]. Since cultural values show the desirable end states that ought to be pursued [22,23,24] they tend to shape members’ attentiveness to or prioritisation of stressors in relation to appraisal processes. Individuals in this study are from the same nation and are socialised to use their culture-specific orientations to guide their daily coping processes. For example, in China where the culture values collectivism (vs. individualism), people tend to form an interdependent (vs. independent) self-construal [25] and prefer to use ought self (vs. ideal self) to guide their behaviours. Obeying government rules on limiting travel, most Chinese employees work at home and schedule times to buy products in the supermarket. Buying products in a supermarket is an opportunity to release their stress during this pandemic. Unneeded consumption often occurs because consumers consider unneeded products can be potentially effective for their family members using interdependent value. Furthermore, an unneeded consumption process can be a positive recovery factor to help release stress after nationwide lockdown and individual insolation. According to the lockdown rules in China, the government permits only one person from a family to buy products outside of the home, therefore, it can be difficult to know all needs of their family members. Thus, unneeded products may be purchased to satisfy family members’ potential needs.

Unneeded consumption behaviour is considered “redundant shopping” [26]. Why do consumers expend their economic resources to buy unnecessary products? When these consumers engage in unneeded consumption behaviour, they may improve their mood [20] by satisfying the psychological needs that the acquisition of necessities might not meet [27]. Such unneeded consumption behaviour fluctuates daily and such fluctuations always coincide with changes in resource conservation and generation [28]. Thus, the conservation of resources theory (COR) may explain the proximal actor-based effects of daily, unneeded consumption behaviour. The COR describes how individuals strive to retain, acquire and attain resources and decrease the threat of net resource loss. In particular, according to COR, people tend to minimise net loss when they are confronted with stress [29]. The theory reveals the regulated process of resources according to the behavioural stress they are experiencing. Stress refers to the reaction to the environment where there is a threat of resource loss, net resource loss, or a lack of opportunities to gain resources; resources include energy, conditions, personal characteristics and anything needed to attain goals [30,31]. A primary understanding of resource loss promotes the idea that losing direct resources is more harmful than gaining the resources that were lost [31].

The present study focuses on the proximal positive effects of unneeded consumption behaviour for actors. From the perspective of resource conservation, there are certain reasons why unneeded consumption behaviour may help consumers avoid loss. Many utilitarian consumption behaviour activities include controlling resources to satisfy basic needs [32]. For example, consumers purchase necessary products at first and buy unnecessary products through buying inertia if they do not control their budget and purchasing motivation. Acts of self-control require more effort, inhibition and stress on limited resources [33,34]. When purchasing intention is induced by certain cues, like cuteness, consumers must expend resources to suppress or inhibit unneeded consumption behaviour, such that the behaviour of self-control will spend their resources with the reminders of responsibility [1]. In other words, suppressing the intention for unneeded consumption behaviour would entail further resource loss for consumers [33]. Engaging in uncontrolled acts like unneeded consumption behaviour, therefore, releases consumers from the resource-consuming situation of behavioural inhibition, leading to avoidance of further resource loss (suppressing their buying motivation and causing the loss of energy) and holding onto their current recovery (releasing their stress caused by COVID-19).

Psychological recovery is defined as the period when people return to a normal mode of functioning by removing related stressors [35]. Recovery processes that often occur during vacations can bring relief from negative emotions in life or work [18]. One of the relaxing choices in vacations is purchasing behaviour. Besides satisfying consumers’ basic needs, the purchasing process helps them relieve stress by the psychological satisfaction of fulfilling their purchasing impulses. Consumers often can judge whether purchasing behaviour is unneeded after the completion of the consumption process. However, during the purchasing process, it is difficult to recognise redundant shopping because consumers are often immersed in emotional experiences. For example, regarding delicious food, consumers may buy more than they need. When consumers buy more bread than they need, they experience happiness and forget their specific needs at that time.

At the very least, consumers pay attention to the satisfaction derived from purchasing their favourite commodities and neglect whether the quantities they buy outweigh their daily needs. In the short term, consumers tend to focus on psychological needs like hedonic value or positive emotions because they do not have enough time to rationalise their purchasing decision (e.g., whether their purchasing behaviour is overconsumption) [36].

Effecting compliance with their psychological needs can replenish consumers’ resources via satisfying basic needs for control [13]. Fritz et al. [37] concluded that there is a beneficial relationship between control perceptions and recovery levels. Additionally, low self-control consumers tend to be motivated to enjoy short-term pleasures, as opposed to high self-control consumers [38]. As a kind of low self-control behaviour, Qin et al. [39] found that abusing others could improve the level of recovery from stressors. In conclusion, the current study suggests that unneeded consumption behaviour might enhance consumers’ recovery levels by preventing further resource loss with beneficial self-control and acquiring new resources by improving their sense of relaxation. Thus, we suggest that beneficial control with unneeded consumption behaviour is good for their recovery and propose the following hypothesis:

**Hypothesis** **H1.**
*Engaging in unneeded consumption behaviour is positively related to consumers’ own recovery level.*


The degree of work engagement fluctuates from one day to another according to individuals’ resources differences [18]. For instance, individuals with high work engagement have high levels of energy, dedication and absorption in the workplace [40,41], and high work engagement improves well-being and in-role or extra-role work performance [40,42]. People with high recovery levels are more resilient, even if they face stress and tend to concentrate on their tasks at work and ignore irrelevant cues [18,43]. Therefore, a resource-rich person is full of energy and has enough resources to draw upon, therefore, tends to be more dedicated and concentrate on their task at work. The present study suggests that people with high levels of recovery might enhance employees’ work engagement by having enough resources to give them energy and the ability to concentrate on their tasks at work. Previous studies have proved this positive influence of recovery level on work engagement [18,41,44].

Unneeded consumption behaviour also influences work engagement. The current study suggests that the recovery level caused by unneeded consumption behaviour aids consumers’ work engagement or investment of physical and psychological energy when they shift enjoyment of consumption to their workplace [43,45]. Unneeded consumption behaviour affects work engagement via recovery level. First, work engagement requires additional personal effort. After unneeded consumption behaviour, sufficient resources are available to concentrate on the task at work. Overconsumption behaviour is very important for consumers to experience positive emotions and employees are willing to expend effort at work [17,46]. Effort expenditure at work can result in strain, whereas during unneeded consumption behaviour, they recover from the previous strain and return to a more relaxed state of feeling refreshed and replenished [47].

Work engagement will benefit from unneeded consumption behaviour. As a result of consumers’ unneeded consumption behaviour, individuals wilfully obtain resources needed for high work engagement. Furthermore, recovery levels will also have an impact on work engagement. Additionally, recovery levels play a crucial role in mediating the effect of unneeded consumption behaviour on work engagement. We, therefore, hypothesise the following:

**Hypothesis** **H2.**
*Unneeded consumption behaviour plays an indirect positive role in work engagement through recovery level.*


### 2.1. Personal and Situational Limitations on the Benefits of Unneeded Consumption Behaviour

The mechanism discussed in the above paragraphs—resource recovery—explains how abusive behaviour might aid consumers’ work engagement. In the present section, a new question will be discussed from the resource perspective. When unneeded consumption behaviour is beneficial for consumers’ engagement in work, COR theory proposes that personal and situational factors constrain people’s reactions to the procedures of decreasing net loss and acquiring new resources. Specifically, individual characteristics and environmental factors that engender additional stress after events and reflect these levels of resources are related to resource conservation and gaining processes and might have implications for the resource-related outcomes of unneeded consumption behaviour [29,30]. Importantly, COR theory states that coping events will be beneficial so long as they create no additional stress for actors. However, unneeded consumption behaviour may create additional stress when people perceive the detrimental outcomes to others’ well-being, as it violates social environment norms and engenders harm to other people. This additional stress might negate the potential recovery effect from unneeded consumption behaviour.

Additionally, the primacy of loss suggests that the gains of recovery are contingent on whether plentiful resources are available [30,48], and resource acquisitions show greater meaning in situations of scarce resources [48,49]. When consumers find they are in resource-scarce situations (e.g., consumers with high indulgence), the positive impact of unneeded consumption behaviour on recovery can be further strengthened. Accordingly, consumers’ perceived consumer effectiveness and indulgence may moderate these recovery processes triggered by unneeded consumption behaviour. In the following section, we explain how these moderators can function in these processes.

#### Moderating Role of Perceived Consumer Effectiveness

Perceived consumer effectiveness is understood as the idea that self-belief toward individual consumption behaviours can play an effective role in protecting resources [50,51]. Additionally, the level of perceived consumer effectiveness is measured as a judgement of themselves in circumstances from the perspective of the related resources [52,53]. Current studies found that perceived consumer effectiveness is more effective than other indicators, such as environmental concern, green product attitude, or knowledge, to predict environmentally sustainable behaviour [53,54,55], which is important for capturing the desired outcomes of green product purchase [56]. Perceived consumer effectiveness is an environment-oriented motivation that aims to improve the well-being of residents and involves sensitivity to environmental desires.

During the COVID-19 crisis, people tended to conserve the natural environment against more virus infections. In collectivistic cultures [57] such as the Chinese culture, consumers will make purchasing decisions with the motivation of protecting others’ health. Unneeded consumption behaviour may damage the environment due to more disposable packages or other harmful materials. In this pandemic, many cities were on mandatory lockdown enforced by the government and necessary goods were very limited. Someone may buy more unneeded products, which may, in turn, be necessary for others. Although unneeded consumption behaviour can aid the recovery process for consumers, such behaviour can also reduce social resources and violate the self-belief of highly perceived consumer effectiveness. Such a violation might cause discomfort or additional stress for consumers as it threatens their good self-image of being a moral representative [58].

While unneeded consumption behaviour has a negative effect on the environment [2], for higher perceived effectiveness consumers, unneeded consumption behaviour violates their ingrained tendency [50], which demonstrates the uncontrolled nature of their behaviour. This decreased sense of control can weaken certain effects of positive emotions, which may be experienced from the enjoyment of unneeded consumption behaviour. However, unneeded consumption behaviour might aid recovery by converting consumers from engaging in resource-consuming situations of self-control to consumers with high-perceived consumer effectiveness, but some additional stress engendered by unneeded consumption behaviour may weaken the gains for recovery [29].

Moreover, high-perceived effective consumers are more likely to protect the environment [56]. Thus, for high-perceived effectiveness consumers, unneeded consumption behaviour tends to violate their ingrained tendency, demonstrating the uncontrolled nature of their behaviour. This decreased sense of control can weaken some recovery effects that may be experienced from the enjoyment of unneeded consumption behaviour. Combining such a relationship with the mediating effect of unneeded consumption behaviour on work engagement through recovery, the present study suggests that perceived consumer effectiveness may attenuate such indirect beneficial effects because recovery tends to lead to gains for resources and enjoying positive emotions in work engagement. Therefore, we propose the following hypothesis:

**Hypothesis** **H3.**
*Perceived consumer effectiveness can moderate the indirect effect of unneeded consumption behaviour on work engagement through recovery level, such that the indirect effect is negatively related to perceived consumer effectiveness.*


### 2.2. Moderating Role of Indulgence

Indulgence captures the extent to which societies allow or promote the gratification related to natural human drives, enjoying life and having fun [59]. The relationship between indulgence and environmental concerns is one of low restraint [60] and it is in this low self-control that unneeded consumption behaviour impacts the recovery level.

Situational characteristics might influence how consumers react to their own unneeded consumption behaviour. Indulgence requires decreased self-control and is often caused by initial resource loss if they are in resource-consuming situations such as incidental damage, life distress or job stress. Consumers tend to indulge and spend when they have made some prepayment of resources, such as money and time [61]. Indulgence is often caused by initial resource loss; consumers will choose indulgence when they confront stress, such as incidental sadness. Thus, consumers high in indulgence experience resource-scarce situations, strengthening the influence of unneeded consumption behaviour on recovery. In particular, unneeded consumption behaviour might improve recovery levels by releasing consumers from such resource-consuming situations of suppressing consumption impulses.

Furthermore, high indulgence demonstrates the opposite situation with regard to self-control. Self-control is required to allocate more resources and energy to balance short and long-term desires [62]. Contrarily, it is easier to consume in high indulgence to satisfy current desires while maintaining sufficient resources because engaging in indulgence can improve consumers’ sense of pleasure or happiness [63]. Low self-control consumers would enjoy their consumption experiences when they indulge for no reason [13]. Therefore, there are fewer recovery-based benefits through unneeded consumption behaviour when suppressing indulgence. Consumers can strengthen and repair their mood by indulging in unneeded consumption behaviours and attain more resources for work, based on the recovery level [20]. Hence, we argue that there should be a correlation between indulgence and unneeded consumption behaviour on work engagement. Therefore, the following is proposed:

**Hypothesis** **H4.**
*Indulgence can moderate the positive indirect effect of unneeded consumption behaviour on work engagement through recovery level, such that the direct effect is positively related to indulgence.*


A theoretical framework was built, based on the mentioned hypotheses, to explore the positive influence of unneeded consumption behaviour on work engagement via the shift of recovery level from life to the workplace as shown in Figure 1.

## 3. Study 1

In study 1, we tested the relationship between unneeded consumption behaviour and consumers’ recovery levels e.g., test H1 by adapting a 2-sided (experiment vs. control) between-subject design.

### 3.1. Samples and Procedure

We chose 100 MBA students as samples from a university in Eastern China because they often shift their roles from shopping centre to the workplace, which fit the research object i.e., the shift from consumption behaviour to work state. There are no conflict of interest using these 100 MBA students. We obtained written informed consent from each of the participants. Among these students, 37% were male and 63% were female; the average age was 26.9 (SD = 4.1) and the average number of years of work experience was 5.3 (SD = 1.8). Most importantly, the average Sunday expenditure among the participants was 158.9 Yuan (about 22.7 USD). Each participant would receive 30 Yuan (about 4.3 USD) if they completed the experimental task. All participants were randomly divided into two groups (experimental group, *n* = 49; control group, *n* = 51). Before we assigned the experimental task, all participants were emailed a link to an online survey administered in Qualtrics.

The task consisted of two procedures. First, a 500 Yuan (about 71.4 USD; more than triple average expenditure among participants—158.9 Yuan, or about 22.7 USD) voucher for a supermarket was assigned to the participants of the experimental group and control group on Saturday. The experimental group participants were required to buy only essential foods such as rice, bread, or meat by spending the full amount of the voucher before 9:00 PM the next day (Sunday). The control group participants were also required to buy only essential foods before 9:00 PM on Sunday, but any remaining value of the voucher could be used in the future. It was specified that the foods purchased by using the voucher were only to be used by themselves. After the participants completed the assigned task, an unneeded consumption behaviour scale had to be completed immediately (a manipulation check). Second, between 9:00 and 10:00 PM on Sunday, all participants were emailed and asked to complete the recovery scale items. After Qualtrics data managers collected all the participants’ data, the vouchers assigned to the experimental group were spent on basic foods.

### 3.2. Measures

The participants responded to these items on a 5-point Likert scale (1 = “Strongly disagree” and 5 = “Strongly agree”). The author used a translation-back translation procedure from English to Chinese. The recovery level was tested by adapting a 3-item scale from Sonnentag [18] (2003; α = 0.70), with items such as: “Because of the shopping activities pursued yesterday, I feel relaxed.”

For the manipulation check, the participants assessed the extent to which they engaged in unneeded consumption behaviour. We adapted Bulut et al.’s 5-item version [26] of the unneeded consumption behaviour scale (α = 0.81); a sample item is: “In this shopping process, I bought products that were not in my mind or shopping list.”

### 3.3. Results and Discussion

#### 3.3.1. Manipulation Check

The *t*-test results demonstrated that participants in the unneeded consumption behaviour condition (M = 3.45, SD = 0.79) rated their unneeded consumption behaviour higher than those in the control condition (M = 1.99, SD = 0.57), t (98) = 6.21, *p* < 0.001, Cohen’s d = 1.36). Based on these results, the experiment manipulation is effective.

#### 3.3.2. Hypotheses Tests

H1 indicates that unneeded consumption behaviour has a positive effect on consumers’ recovery level. The results from the ANOVA demonstrated that the participants in the unneeded consumption behaviour group (M = 3.73, SD = 0.70) presented significantly higher values of recovery than those in the control group (M = 3.26, SD = 0.66, F (1, 98) = 6.88, *p* < 0.05, η2 = 0.12), thus supporting H1.

#### 3.3.3. Discussion

The findings of study 1 reveal that consumers can attain higher levels of recovery with the help of unneeded consumption behaviour. However, this experiment still has certain limitations. First, although the values of the vouchers assigned to the participants were much higher than their average expenditure on Sunday, the basic expenditure of a few participants may be more than 500 Yuan on different days of the week (including Sundays), meaning that they cannot continually experience the process of unneeded consumption behaviour. It would therefore be better to examine whether these participants react similarly on different days of the week.

Second, although these theoretical hypotheses are not situation-specific, the study examined the direct relationship between unneeded consumption behaviour and recovery. We do not know at what point the beneficial effect of unneeded consumption behaviour on recovery level can be generalised to other situations; perhaps such an effect is more likely to emerge in high indulgence situations, like at amusement parks, where consumers may believe that redundant consumption is dispensable. To redress these limitations, study 2 conducted daily multi-wave diary questionnaires to explore the moderating factors and establish external validity on the proposed full model by a field design.

## 4. Study 2

In study 2, we tested the mediating effect of the recovery level between unneeded consumption behaviour and work engagement, which was moderated by consumer indulgence and perceived consumer effectiveness.

### 4.1. Samples and Procedure

The experience sampling method was employed in this study to capture the natural state change of redundant purchase behaviour and recovery at short intervals to reveal the causal relationship accurately [64]. To obtain representatives who engage in both unneeded consumption behaviour and work, we collected data from 150 consumers from China who were employed by at least one company during the COVID-19 pandemic. We recruited participants in a shopping mall and they were selected based on two standards: working from home and being in lockdown management by the resident committee.

There were two types of questionnaires: a fundamental questionnaire and a daily questionnaire. First, the fundamental questionnaire was used to document participants’ information, including demographic information, work engagement and their levels of indulgence and perceived consumer effectiveness at the inter-personal level. A week later, the daily questionnaire was used at three points in time [65] and intra-personal variables such as unneeded consumption behaviour, recovery and work engagement were included. At 8:00 PM on Sunday, we collected the data about unneeded consumption behaviour. Then at 8:00 AM and 11:00 AM of the following Monday, no more than 15 h after time 1 of the previous Sunday, we measured the levels of recovery and work engagement, respectively. We chose Sunday evening for time 1, Monday morning for time 2 and noon on Monday for time 3 because Sunday is a non-work day and consumers can fully experience their consumption.

Furthermore, the following Monday is the first day of the work week, meaning that it is possible to capture the unneeded consumption behaviour in the change process from consumption to work. We collected data over ten iterations of the two days, from Sunday to the following Monday. Only 115 fundamental questionnaires from the initial 150 candidates and 1150 (115 × 10) daily questionnaires were received over 11 weeks. In the effective samples, 42.61% of participants were female, the average age was 30.18 (standard deviation 5.25). The average education level of all participants was 2.82 (standard deviation 0.57).

### 4.2. Variable Measurement

All the measurement scales applied were from studies written in English; therefore, we used a Chinese-English back translation procedure to ensure that the translated content conformed to the original meaning in English [66]. All scale items were reported by participants with a 5-point Likert scale (1 = “strongly disagree” and 5 = “strongly agree”). In the daily questionnaires, we added “today” as the introduction description before the participants completed the questionnaires.

For unneeded consumption behaviour (Table 1), we used a 5-item scale, including items such as “I buy new products even if I own similar ones.” The average Cronbach’s α across 10 days = 0.88, average variance extracted (AVE) = 0.58 and composite reliability (CR) = 0.89. For the recovery level (Table 2), we used a 3-item scale adapted by Sonnentag et al. [28], including items such as “I have felt relaxed.” The averaged Cronbach’s α across 10 days = 0.87, average variance extracted (AVE) = 0.60 and composite reliability (CR) = 0.89.

For work engagement (T3), we used a 3-item scale used by Lanaj et al. [67] and included items such as “I was immersed in my work.” The Cronbach’s α across 10 days = 0.79, average variance extracted (AVE) = 0.53 and composite reliability (CR) = 0.82.

For perceived consumer effectiveness, we used a 4-item scale developed by Kim and Choi [52], including items such as “Each person’s behaviour can have a positive effect on society by signing an appeal in support of promoting the environment.” The Cronbach’s α = 0.80, average variance extracted (AVE) = 0.57 and composite reliability (CR) = 0.83.

For indulgence, we used scenarios adapted from Suzuki et al. [14]. All participants were required to answer a question about whether they had made a huge mistake, meaning that, products they bought were unnecessary. Only 115 of 150 participants answered the question, which confirmed unneeded consumption behaviour situations and demonstrated their willingness to engage in mood-repairing indulgence during the COVID-19 pandemic. Indulgence was measured with the item “you buy to engage in indulgent consumption under the situation above.”

A 3-item scale was used for trait work engagement, adapted from the Utrecht Work Enthusiasm Scale (UWES) by Schaufeli et al. [45]. These items cover vigour, dedication and absorption, and the scale includes items such as “I feel strong and vigorous in my work.”

### 4.3. Analytic Strategy

First, we primarily proved the hypotheses with a correlational analysis. Then, using a confirmatory factor analysis (CFA), we tested the discriminant validity of the key variables of the research model, including the between-personal variables (perceived consumer effectiveness, indulgence and trait work engagement) and within-personal variables (unneeded consumption behaviour, recovery level and work engagement). Considering the within-personal data nested in the between-personal data, we used multilevel structural equation modelling to test the hypotheses of the moderated mediating model in M-plus 7.0. When testing the moderated cross-level effect, we processed the data using centring by group mean on the within-personal level and by total mean on the between-personal level to avoid the influence of a false cross-level effect [68]. Above all, multilevel structural equation modelling can test the direct and mediating effects (H1 and H2).

Subsequently, a parametric bootstrap was used to calculate the significance of the indirect effects [69]. A Monte Carlo simulation with 5000 replications was used to assess the confidence interval (CI) and we calculated the value of indirect effects in accordance with this formula [70]. Thus, this method can test the moderated mediating effect (H3 and H4), such that it is possible to examine the mediating effect of recovery at different levels of perceived consumer effectiveness or indulgence (above or below one standard deviation). On the between-personal level, democratic variables such as sex, age and education were controlled. On the within-personal level, we controlled necessary consumption expenditure per week and income per week (the highest expenditure or income in this survey was no more than 5000 Yuan) and money values were transformed to a scale ranging from 1 to 5 (with “1” representing 1000 Yuan) because democratic variables, necessary consumption and income could have an important influence on recovery or work engagement [71].

### 4.4. Results and Discussion

#### 4.4.1. Correlation Analysis and CFA

Table 1 indicates that demographic variables such as sex, age, education and necessary consumption expenditure were not related to key variables like indulgence and recovery level; hence, they were excluded from the multilevel structural equation modelling in the next step. Additionally, income was positively related to recovery level (r = 0.09, *p* < 0.05). On the within-personal level, unneeded consumption behaviour was positively related to recovery level (r = 0.33, *p* < 0.001) and recovery level was positively related to work engagement (r = 0.27, *p* < 0.01). Meanwhile, unneeded consumption behaviour was positively related to work engagement (r = 0.39, *p* < 0.001). These results primarily supported H1 and H2. Notably, income was positively related to some key variables (unneeded consumption behaviour and recovery level) and had to be included in the multilevel structural equation modelling.

On the between-personal level, the results from the CFA indicated that the two-factor model (perceived consumer effectiveness and indulgence) attained a more ideal state (χ2/df = 1.79, CFI = 0.97, RMSEA = 0.05) than the one-factor model combining perceived consumer effectiveness and indulgence (χ2/df = 3.55, CFI = 0.80, RMSEA = 0.13). On the within-personal level, the results from the CFA demonstrated that the three-factor model (unneeded consumption behaviour, recovery level and work engagement) achieved a more ideal state (χ2/df = 1.12, CFI = 0.98, RMSEA = 0.03) than the two-factor model collapsing recovery level, work engagement (χ2/df = 4.86, CFI = 0.76, RMSEA = 0.16) and other models. Consequently, the two-factor model on the between-personal level and the three-factor model on the within-personal level demonstrated satisfactory discriminant validity.

#### 4.4.2. Hypotheses Tests

As illustrated in Table 2, the results indicated a significant positive relationship between unneeded consumption behaviour and recovery level after adding a control variable (β = 0.30, SE = 0.08, *p* < 0.001); recovery level was positively associated with work engagement (β = 0.28, SE = 0.09, *p* < 0.01), supporting H1 and H2.

The interacting effect on recovery level was tested in Table 2, Figure 2a,b. The moderating role of perceived consumer effectiveness on the relationship between unneeded consumption behaviour and recovery level was significantly negative (β = –0.39, SE = 0.18, *p* < 0.05) and the relationship between unneeded consumption behaviour and recovery level was heightened when perceived consumer effectiveness was low (M-SD), partly supporting H3. The moderating role of indulgence on the relationship between unneeded consumption behaviour and recovery level was significantly positive (β = 0.35, SE = 0.18, *p* < 0.05), and the relationship between unneeded consumption behaviour and recovery level was heightened when indulgence was high (M+SD), partly supporting H4. In particular, the relationship between unneeded consumption behaviour and recovery level switched from a negative association to a positive association when indulgence was high (M+SD). Similarly, this relationship switched to a negative association when indulgence was low (M-SD).

Furthermore, the results indicate a moderated effect of perceived consumer effectiveness on the indirect effect of recovery level. This indirect effect was proved (γ = 0.11, 95% CI = [0.02, 0.13], excluding zero) to occur when perceived consumer effectiveness was low (M-SD), but the indirect effect was not significant (γ = –0.01, 95% CI = [–0.03, 0.01], including zero) when perceived consumer effectiveness was high (M+SD). Furthermore, the significant differentiation between high (M+SD) and low (M-SD) was proved (γ = 0.12, 95% CI = [0.05, 0.13], excluding zero), supporting H3. Meanwhile, the results supported H4. The indirect effect was significant (γ = 0.15, 95% CI = [0.05, 0.18], excluding zero) when indulgence was high (M+SD), but it was not significant (γ = –0.02, 95% CI = [–0.04, 0.02], including zero) when indulgence was low (M-SD). Furthermore, the differentiation between high (M+SD) and low (M-SD) was significant (γ = 0.16, 95% CI = [0.08, 0.17], excluding zero).

## 5. Conclusion and Implications

### 5.1. Conclusions

The present research introduces a primary trial to discern the benefits of unneeded consumption behaviour to consumers. The shift from consumption to the workplace is especially highlighted with the proximal benefits of unneeded consumption behaviour for actors, including recovery and work engagement. Such beneficial effects can be weakened or strengthened by perceived consumer effectiveness and indulgence, respectively. The findings of short-term benefits (no more than 15 h) can be used to effectively control the frequencies of unneeded consumption behaviour through interventions that help consumers recover at the workplace or change the levels of perceived consumer effectiveness and indulgence. This study may contribute to the trend of exploring how various consumption behaviours influence consumers themselves.

### 5.2. Implications

The current study makes certain theoretical contributions. First, this study broadens the knowledge of unneeded consumption behaviour on the within-personal level. Although previous research has explored various outcomes of unneeded consumption behaviour, the impact of unneeded consumption behaviour on consumers has largely been overlooked. Furthermore, previous studies have focused on the damage or detrimental impacts of overconsumption behaviour, including often cost to consumers and the environment [72]. However, this study captured the short-term benefits of unneeded consumption behaviour on consumers. To support our hypothesis, we emphasised the finding that unneeded consumption behaviour plays a positive role in work engagement via recovery level.

Second, this study contributes to the consumption theory by illustrating when unneeded consumption behaviour can reveal certain benefits (recovery and work engagement) for consumers. To completely understand the effects of unneeded consumption behaviour on consumers, it is necessary to find the boundary conditions when unneeded consumption behaviour influences actors in weaker or stronger ways. In the perspective of COR, these conditions are revealed both individually (indulgence) and contextually (perceived consumer effectiveness). In particular, unneeded consumption behaviour is beneficial for recovery and engagement at the workplace when consumers have low perceived consumer effectiveness. If consumers have high perceived consumer effectiveness, unneeded consumption behaviour will engender additional resource loss, which weakens the benefits of the recovery and replenishing processes. Meanwhile, the conserving effect of resources also hinges on the extent of indulgence. The benefits of unneeded consumption behaviour on recovery or engagement were stronger when consumers had high indulgence levels.

In practice, these findings may make several contributions. The key findings provide a possible reason why consumers engage in unneeded consumption behaviour. Unneeded consumption behaviour may help consumers conserve their resources by freeing them from resource-consuming situations under which they must control and suppress their impulses. Such resource conservation and gain can shift from consumption to the workplace through work engagement. Unneeded consumption behaviour is not the first order of resource recovery given the harmful effects of unneeded consumption behaviour on the environment. For example, consumers can enjoy tourism as a kind of leisure activity.

The current findings revealed that unneeded consumption behaviour could cause some sense of guilt for consumers with high-perceived consumer effectiveness. These consumers will experience the discomfort of unneeded consumption behaviour when the behaviour harms the environment. Thus, it is easy to control unneeded consumption behaviour by imparting environmental knowledge and emphasising the effectiveness of overconsumption for consumers.

Finally, the current study demonstrated that consumers’ perceptions of indulgence could strengthen the relationship between unneeded consumption behaviour and recovery. Another tentative way to control unneeded consumption behaviour is to alleviate consumers’ perceptions of high indulgence. For example, consumers experiencing greater consumption happiness can be more satisfied with subsequent indulgence [13]. Some values given to indulgence, such as religiousness, can influence their experiencing happiness [60]. Thus, we can broadcast green values for consumers to reduce their sense of high indulgence and control the occurrences of unneeded consumption behaviour.

### 5.3. Limitations and Directions for Future Research

Although the current study has some implications for theory and practice, some limitations and research directions for the future must be addressed.

First, based on the actor-centric effects of unneeded consumption behaviour, study 2 collected self-reported data of daily unneeded consumption behaviour, recovery and work engagement. Thus, it can cause the limitations of common method variance (CMV) [73]. However, based on the suggestions from Ilies et al. [65], we used three points in time to collect the data of unneeded consumption behaviour, recovery and engagement, respectively, to reduce the level of CMV.

Second, we tested the moderating effects of indulgence and perceived consumer effectiveness, capturing the influence of consumers’ individual and contextual factors on the shifting relationship from unneeded consumption behaviour to engagement in the workplace. In addition to these factors, future research can explore the impact of a team or organisational factors. For example, team psychological safety may provide a positive context for recovering from some stressors [74], so that consumers can recover, with no sense of guilt, from the detrimental effects of unneeded consumption behaviour.

Third, study 2 examined the effects of unneeded consumption behaviour on recovery and engagement within only 15 h, meaning that the study only captured the short-term benefits for consumers. In the long run, consumers may attain different emotions or satisfaction when they reappraise the unnecessary or indulgent behaviour over time [11]. Thus, future research should focus on the long-term effects of unneeded consumption behaviour.

Finally, the current study used a sample only from China, during the COVID-19 pandemic, and these sample sizes were limited because of the difficulty in recruiting participants because of the lockdown policy. Samples from other countries were not used for this same reason. Cultural differences may influence how consumers appraise the value of unneeded consumption behaviour. For example, with the value of economic orientation, unneeded consumption behaviour is a positive means to promote the quantities of consumption and indirectly evoke economic development. However, under the belief of social orientation, unneeded consumption behaviour must be forbidden because it can damage the environment and be harmful to subsequent generations. Thus, an intriguing direction for scholars to focus on would be on possible cultural contingencies of unneeded consumption behaviour. For example, with peaks at Christmas, Easter and Black Friday, the quantities of consumption will increase which is not caused by disaster-related panic event [9]. Future research must control the influence of the Chinese Spring festival in COVID-19 on unneeded consumption behaviour and test the recovery level of such behaviour more effectively.

## Figures and Tables

**Figure 1 ijerph-18-06404-f001:**
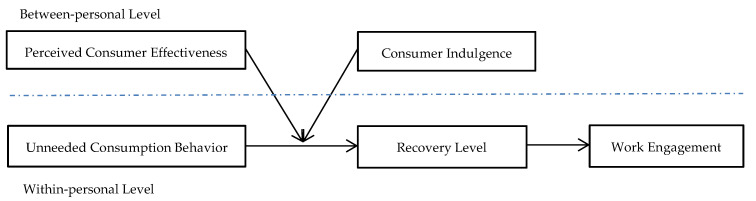
Theoretical Framework.

**Figure 2 ijerph-18-06404-f002:**
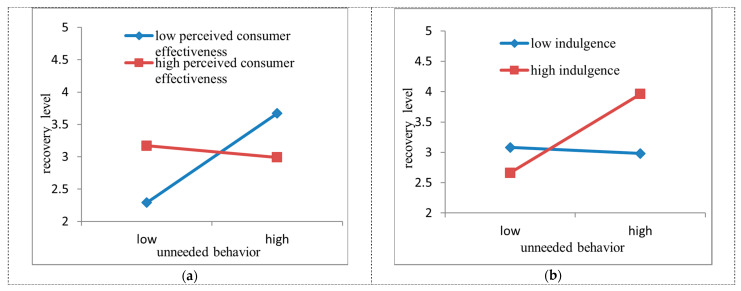
Moderation Plots. (**a**) perceived consumer effectiveness, (**b**) indulgence.

**Table 1 ijerph-18-06404-t001:** Descriptive Statistics, correlations and reliability.

**Between-Personal Variable**	**M**	**SD**	**1**	**2**	**3**	**4**
1. Sex	0.43	0.12				
2. Age	30.18	5.25	0.03			
3. Education	2.82	0.57	0.06	0.02		
4. Perceived consumer effectiveness	3.34	0.39	0.04	0.14	0.17	(0.80)
5. Indulgence	3.59	0.29	0.07	0.10	0.19	0.14
**Within-personal Variable**	**M**	**SD**	**1**	**2**	**3**	**4**
1. Necessary consumption expenditure	1.15	0.16				
2. Unneeded consumption behaviour	3.03	0.41	0.04	(0.88)		
3. Recovery level	3.17	0.39	0.05	0.33 ***	(0.87)	
4. Work engagement	3.33	0.29	0.07	0.39 ***	0.27 **	(0.79)
5. Income	2.44	0.79	0.06	0.11 *	0.09 *	0.06

Note: M = mean; SD = standard deviation; in the between-personal level, *n* = 115; in the within-personal level, *n* = 1150; the data in brackets are reliability, *** *p* < 0.001, ** *p* < 0.01, * *p* < 0.05, two-tailed.

**Table 2 ijerph-18-06404-t002:** MSEM Results.

Variables	Recovery Level		Work Engagement	
	Estimate	*SE*	Estimate	*SE*
*Control variable*				
Income	0.12 *	0.07	0.08	0.07
*Independent variables*				
Unneeded consumption behaviour	0.30 ***	0.08	0.17 *	0.09
Moderator				
Perceived consumer effectiveness	0.05	0.11		
Indulgence	0.14	0.10		
*Mediators*				
Recovery level			0.28 **	0.09
*Interaction terms*				
Unneeded consumption behaviour×Perceived consumer effectiveness	−0.39 *	0.18		
Unneeded consumption behaviour × Indulgence	0.35 *	0.18		

Note: In the between-personal level, *n* = 115; in the within-personal level, *n* = 1150; standardised coefficients were reported. SE = Standard error. * *p* < 0.05, ** *p* < 0.01, *** *p* < 0.001, two-tailed.

## Data Availability

Data available on request due to restrictions i.e., privacy.

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
