# Peer review of "The Positive Effects of Unneeded Consumption Behaviour on Consumers during the COVID-19 Pandemic"

_ijerph, 2021, doi:10.3390/ijerph18126404_

Round 1
Reviewer 1 Report
Figure 2 has some duplicate charts (with an extra line). You might want to delete the duplicates.
Author Response
We are sorry to make this mistake on figure formatting, and have modified the extra line with deleting the duplicate chart. Thank you for your suggestions.
Reviewer 2 Report
The paper has improved but the relevant literature still needs a broader discussion of what constitutes unneeded consumption. There are several interesting discussions on this issue in some on the literature on COVID-19 and so-called panic buying, which the review in this manuscript does not engage with strongly enough. Apart from providing a review, one recent paper: Hall, C. M.; Fieger, P.; Prayag, G.; & Dyason, D. 2021. "Panic Buying and Consumption Displacement during COVID-19: Evidence from New Zealand" Economies 9, no. 2: 46. https://doi.org/10.3390/economies9020046, for example, notes how the consumption during COVID-19 was comparable to that experienced during festive events, such as Christmas, such observations therefore provide an important reframing of the COVID-9 consumption and the reasons behind it. And provides an important counterpoint for future research to some of the suggestions made in the present manuscript.
Author Response
Thanks for your suggestion.
We add the objected literature in two parts. First, we discern the difference between unneeded consumption and panic buying at motive and behavioral characteristic in the second paragraph on the page 1. Second, in the last paragraph, festival event is really factor which will promote the quantities of consumption. Such influence may be mixed with crisis event. We add related limitation and future direction in the page 15.
This manuscript is a resubmission of an earlier submission. The following is a list of the peer review reports and author responses from that submission.
Round 1
Reviewer 1 Report
Thank you for the opportunity to review. This paper investigates how unneeded consumption behavior affect consumers during COVID-19. I enjoy reading the paper. Below are my comments. Hope they are helpful.
- P1 line 33: What is the definition of unnecessary consumption? Is it buying too much that are beyond what one can consumer (e.g., buying too much bread)? Is buying a Hermes bag considered as unnecessary consumption? Or is it buying duplicated products? Please be clear and provide a definition in the introduction.
- P1 line 38: Why is disposal of packaging material social problems? Should it be environmental problems?
- P6 Figure 1: What is the difference between “perceived consumer” and “consumer”? This figure is confusing in the between-personal part.
- P6 Experiment 1: Have you specified which supermarket to use? Did all participants use the same supermarket? 500 Yuan is too much for rice, bread, or meat in some supermarkets. But participants can easily spend it in other supermarkets such as Costco or Metro where the packages are very big.
- P10 Table 1: Why are the correlations of perceived consumer effectiveness with itself 4 is -0.8? Should it be 1? Similar questions between 2. Unneeded consumption behavior and itself….
- P11 Figure 2: Could you please write which variable is x-axis and which variable is y-axis in (a) and (b)?
Author Response
We appreciate for your detailed comments.
And that is a significant contribution to the success of this manuscript, all the problems that the professor mentioned I have completed as below.
- We added the definition of unneeded consumption behaivior. Unnecessary consumption is a kind of state when consumers buy more products than their necessary quantities. Hence, we can simply understand, if at first you need to buy 1 bread, but when you go to a restaurant, because they recommend a promotion, you decide to buy 2 bread, this is also considered unnecessary consumption. It is different from conspicuous consumption that spending of money on and the acquiring of luxury goods and services to publicly display the economic power of the income or of the accumulated wealth of the buyer. For example, Hermes bag. The definition was showed at P1 line 33-39.
- We modified the "environmental" ploblem as your recommendation and the content was re-aranged clearly at P1 line 41.
- P6 Figure 1: To answer the professor's questions, we can explain the following: "perceived consumer" is source from consumer' estimation subjectively. "consumer" is fit to the fact. For example, one consumer think this model is effective, but the fact may be no effective. Thus, we add "perceived". Otherwise, we use "consumer" directly.
- P6 Experiment 1: We use the same supermarket as long as the other elements are unchanged. That is the need to buy a product with the same volume in the same market.
- P10 Table 1, the data in brackets is reliability of the variable, it's not the correlation.
- P11 Figure 2. Thanks for the professor's prompt, we added axis name on figure 2. It will help the reader better understand.
Thank you for your comments.
Reviewer 2 Report
In the manuscript titled "The positive effects of unneeded consumption behavior on consumers during the COVID-19 pandemic" the issue of consumer behaviour during an ongoing epidemic crisis is undertaken. The authors carried out and described two studies in which they analyse an important aspect of human wellbeing. I mean, coping with the challenging situation that may be in opposition to sustainable behaviour.
The formulated statements and conclusions are adequate to the purpose and scope of the work. The authors underline the benefits of unneeded consumption behaviour but are aware of its detrimental effects. At the same time, the proposal to counteract the harmful effects is given.
Additional comments:
Line 284: To the expresion "unneeded consumption behavior" may be added information, that it is experimental group. A short line about the statistical analysis regarding Study 1 should be added. The description of Study 2 can be enriched with the questions example from the questionnaires. Line 489 The caption under the figure should be more detailed. And axis X and Y labelled.Author Response
Dear Reviewer,
Thank you for your time to consider our manuscript.
- Line 284: To emphasize "unneeded consumption behavior", as on the advice of professors, we added information as an "experimental group" at P7 line 344.
- A staistical analysis for Study 1 was highlighted as following at P8 of Hypotheses Tests sesion title. "The results from the ANOVA demonstrated that the participants in the unneeded consumption behavior group (M = 3.73, SD = .70) presented significantly higher values of recovery than those in the control. group (M = 3.26, SD = .66, F (1, 98) = 6.88, p <.05, η2 = .12), thus supporting hypothesis 1".
- Study 2 at Variable Measurement, to consolidate connectivity and seamlessness, some question example has been noted at P9 line 439-451.
- Finally, Line 489: a short line of axis X and Y was labelled at P11 line 493.
Thank you for your comments.
Reviewer 3 Report
Thank you for the opportunity to review the paper which touches on an important area of consumption behaviour.
The literature review could be substantially improved, with further literature on consumption behaviours related to and crisis and disaster response being noted. Very importantly there is a cultural context to such response which needs to be discussed in this paper and not relegated to an also ran in the final paragraph, i.e. it is something obvious to be noted from the outset because it contextualises the work.
The research model is very weak and is really not as detailed as it should have been. The nature of what constituted unneeded consumption behaviour is not detailed - and the notion of need is not considered enough. For example, was it planned consumption brought forward in time. To ignore the home environment is also poor as the consumption may have served to reinforce family ties and relations in home environment rather than for self. As such the nature of such unneeded consumption really requires greater detail so that the results can be used more positively. This is especially disappointing given that the daily approach adopted in study 2 is a very good idea and is potentially data rich. Indeed, it would be helpful to understand the unneeded consumption in the broader context of the respondents' consumption.
Details of ethical clearances are not provided to readers. This is especially concerning with study 1 in which, if the MBA students were your own students, there are issues of compulsion and, aligned with that, the potential for the data to be distorted. Sample sizes are disappointing, especially for study 2, and also raise issues of their representativeness.
Overall this is a potentially valuable study the contribution of which is greatly limited by the model adopted, the level of detail sought (or at least communicated to readers), and its contextualisation.
Author Response
Dear Reviewer,
We are very grateful that you took your valuable time to review our manuscript. Based on your assessment, we have adjusted some points as below:
- First, to reinforce literature on consumer behavior, we have added cultural response literature and related backgroud in the P2 (from line 91 to line 113). In crisis and disaster response, consumer behavior will choose appropriate reactions and shap the ways members assess or cope with stressors related with the COVID-19 pandemic which was vary by country culture foundation. It is our omission to omit this recognition at the beginning of the manuscript.
- Second, in the home environment, the Covid-19 pandemic caused a lot of pressure on members, especially women, such view contributed to the unneeded consumption behavior, it was seen as a demand to reduce stress. A example was added on P2 (line 47-53).
- Third, related to sample size. We found that sample size was a shortcoming of the study and it was put in the limitations. In addition, to demonstrate data clarity, an ethical clearances statement of responnders was added. MBA students have no interest conflict with all researchers showing in the paper at P7 (line 327-329).
Finally, we would like to extend our sincere thanks to you. Hopefully the above amendments can increase the persuasion of the manuscript. Please refer to the red text of the revised manuscript for your conference.
Thank you for your comments.
Round 2
Reviewer 3 Report
Thank you for your efforts in revising the paper. Unfortunately, the quality of English in the sections you have revised is poor and, at times, difficult to understand what you are conveying.
There is also a need to better describe and explain what unneeded consumption means given that needed/unneeded changes over time and in relation to different factors
You conclude: "the data were collected and limited by the number of samples is not large enough to represent variables". Even allowing for the poor English this then suggests to me that there is not a substantial enough methodological - at present - to publish given that the variables cannot be adequately evaluated. While I appreciate the candour and honesty - and agree with it - this therefore makes it very difficult to accept the paper for a quality journal unless there is some methodological dimension of the study that I am unaware of.